# Association between socioeconomic position and occupational health service utilisation trajectories among young municipal employees in Finland

Hilla Sumanen ![ORCID] ,[1,2,3] Jaakko Harkko ![ORCID] ,[3] Kustaa Piha,[2] Olli Pietilainen,[2] Ossi Rahkonen,[2] Anne Kouvonen ![ORCID] [3,4,5]

[1]Health Care and Emergency Care, South-Eastern Finland University of Applied Sciences, Kotka, Finland
[2]Public Health, University of Helsinki, Helsinki, Finland
[3]Social Sciences, University of Helsinki, Helsinki, Finland
[4]Centre for Public Health, Queen's University Belfast, Belfast, UK
[5]Research Institute of Psychology, SWPS University of Social Sciences and Humanities, Wroclaw, Poland

**Correspondence to**
Dr Hilla Sumanen;
hilla.sumanen@helsinki.fi

## ABSTRACT

**Objectives** To identify groups of municipal employees between the ages of 20 and 34 years with distinct utilisation trajectories of primary care services provided by occupational health service (OHS), measured as the annual number of OHS visits, and to identify demographic and socioeconomic risk factors that distinguish employees in the high utilisation trajectory group(s).

**Methods** The present study is a retrospective register-based cohort study. All municipal employees of the City of Helsinki, Finland, aged 20–34 in the Helsinki Health Study, recruited from 2004 to 2013, with follow-up data for 4 years were included in the study (n=9762). The outcome measure was group-based trajectories of OHS utilisation, identified with a group-based trajectory analysis. The demographic and socioeconomic variables used to predict the outcome were age, first language, educational level and occupational class. The analyses were stratified by gender.

**Results** A large proportion of the young employees do not use OHS. Trajectory groups of 'No visits' (50%), 'Low/increasing' (18%), 'Low/decreasing' (22%) and 'High/recurrent' (10%) use were identified. We found occupational class differences in OHS utilisation patterns showing that lower occupational classes had a higher propensity for 'High/recurrent' OHS utilisation for both genders.

**Conclusions** Preventive measures should be targeted particularly to the trajectory groups of 'Low/increasing' and 'High/recurrent' in order to intervene early. In addition, OHS utilisation should be closely monitored among the two lowest occupational classes. More research with longitudinal OHS data is needed.

## INTRODUCTION

Finland has a unique occupational health service (OHS) system with statutory prevention of occupational health hazards (preventive services) and additionally purchased primary care services. OHS may be provided by employer's own OHS units, private clinics or public sector health centres with specifically trained occupational health physicians, nurses, physiotherapists and psychologists.

### Strengths and limitations of this study

► This is the first study using longitudinal occupational health service data.
► The sample consisted all of 20–34-year-old employees of the City of Helsinki. The occupational health-care policies are same for all these employees and have remained same during the study period of 2004–2017.
► Our study avoids a common limitation of previous occupational health service studies that are based on limited samples with data on only those attending the service or those responding to survey.
► Limitations of the data include the lack of diagnostic information and lifestyle factors.
► The lack of information about other primary care visits outside the occupational health service further limits the interpretation of the results.

In Finland, most employers purchase the additional primary care element for their employees. In 2017, 94% of the employees covered by statutory preventive OHS also had an access to primary care for any illness including all non-work related illnesses, paid by their employer and partly subsidised by the National Pension Fund.[1] OHSs are free for employees at the point of delivery and their accessibility is typically good, making them the main source and the preferred type of primary care for Finnish employees.

The utilisation of OHS in Finland or elsewhere has been only scarcely studied, and especially studies using longitudinal data are lacking. There are few previous Finnish studies concentrating on the OHS primary care visits with cross-sectional study designs and from the viewpoint of service utilisation over the course of 6–12 months. A recent study[2] with data from a large private Finnish OHS provider investigated the top 10% frequent attenders in primary care services

in 2015. The results showed that frequent attendance was associated with female gender, being employed by a medium or a large company, working in the manufacturing industry, public administration or in health and social care services. In an earlier study with survey data (n=1636) from the Finnish working-age population,[3] 57% of Finnish employees covered by the OHS primary care visited either their occupational health physician or nurse due to any illness during the 6 month period. In that study, those visits were strongly associated with chronic illness impacting occupational health and work ability. Both previous studies using Finnish OHS data acknowledge the lack of research focusing to these unique services provided for the working population and identify the need for further study to identify service development needs and possibilities.

The current study uses longitudinal data from the own OHS unit of Finland's largest employer. The City of Helsinki offers same OHSs with primary care for all its employees (n=~38 000 per year) with no cost for the employees. The City of Helsinki employees have been the focus of the Helsinki Health Study (HHS)[4] since 2000, but this is the first study using their OHS data. Our focus is on the younger employees and their OHS utilisation, as previous studies have shown that they have a high prevalence of sickness absence (SA)[5] and there are already large socioeconomic differences in SA apparent in the younger age groups.[6 7] Reducing SA is high on employers' agenda. Service utilisation information is important for planning preventive actions via targeted interventions or improving case management protocols[8 9] arranged by the OHS. According to extensive Finnish and international evidence, socioeconomic differences in SA are large,[10–21] thus it would be important to monitor the differences in OHS utilisation from the socioeconomic viewpoint and to be able to identify groups for interventions.

In our study, we aimed to identify developmental trajectories of OHS primary care service utilisation among 20–34-year-old municipal employees of the City of Helsinki. In the second stage of the analysis, we aimed to identify occupational class differences in belonging to different trajectory groups. We tested two hypothesis, first, that the distinct trajectories can be identified and, second, that occupational class gradient can be found in OHS utilisation.

## METHODS

This is a retrospective register-based cohort study. The study is a part of the HHS on health and well-being among employees of the City of Helsinki, Finland.[4] The study included all Helsinki City employees aged 20–34 at the beginning of their first work contract with the City (n=22 576) between 04/06/2004 and 04/19/2013. The selection of this age group was based on the Eurostat definition of young employees and on previous studies investigating the occurrence of illness and SA in employees of different ages.[22 23] For each employee, the follow-up

started from their initial recruitment. Employees with incomplete data on occupational position (n=754) and those with employment record for less than 4 years (n=12 512) were excluded from the present study and we ended up with a sample size of n=9762. We excluded employees with less than 4 years of employment history, defined as being employed 4 years from the initial recruitment and at least 180 days of employment for each year after the recruitment, as we needed a long enough follow-up time to observe potential development trajectories. The descriptive characteristics of excluded subjects are found in online supplementary web appendices 1 and 2. The length of the follow-up was measured as calendar days in employment.

The outcome of the study was OHS utilisation trajectory. The OHS primary care services offered to the employees have remained same during the whole study period. The trajectory, that is, the developmental course of OHS utilisation, was measured from four consecutive data points indicating annual number of outpatient primary care visits for each employee. The number of visits ranged from 0 to 103. The demographic and socioeconomic variables used to predict the outcome were age, first language, educational level and occupational class. The analyses were stratified by gender.

We used four occupational class groups. Based on the socioeconomic classification of Statistics Finland and the occupational classification of the City of Helsinki,[4 12] non-manual employees were divided into three groups based on skills requirements and supervisory status: managers and professionals, semiprofessionals and routine non-manual employees. Managers have subordinates and they do managerial or administrative work, whereas professionals include employees with a university degree, such as physicians and teachers. Semiprofessionals include occupations such as registered nurses and technicians. Routine non-manual employees include clerical employees and lesser-educated occupations particularly within the social and healthcare, such as child-minders and care workers. The fourth occupational class, manual workers, include occupations for example from the fields of cleaning, kitchen work and public transport.

Age was measured at the beginning of the follow-up and was categorised into three groups: 20–24, 25–29 and 30–34 year-olds. First language was categorised as: Finnish, Swedish and Other. Education was classified into three levels: higher education (a master's or a doctoral degree), upper secondary (a Bachelor's degree) and lower secondary (upper-secondary school, vocational school) or basic education (comprehensive school).

The employer's personnel and occupational healthcare registers were used to obtain sociodemographic characteristics and information on OHS use. Educational level was obtained from annually updated Statistics Finland's registry of completed education and degrees and was linked to the City of Helsinki personnel register.

## Ethics

The study follows the HHS protocol in line with the University of Helsinki's guidelines and data legislation. The ethics committees of the Department of Public Health, the University of Helsinki and the health authorities of the City of Helsinki have approved the HHS study. The City of Helsinki has given permission for data linkage.

## Patient and public involvement statement

No patient or public involvement.

## Statistical methods

Group-based trajectory modelling (GBTM) with Stata's *traj* command[24] was applied to identify clusters of individuals or trajectory groups, with similar developmental trajectory on OHS utilisation. The method assigns a subject to a trajectory group by assessing the probability of group membership. The count variables were assumed to be Poisson-distributed and zero inflated Poisson models were applied. The ideal number of trajectory groups and trajectory shapes were assessed by four criteria suggested by the existing literature: Bayesian information criteria (BIC), posterior probabilities of trajectory group membership higher than 0.70, sizes of trajectory groups larger than 5% and a distinct interpretability of the identified trajectory groups.[25 26] Subsequently, multinomial logistic regression models using Stata's *mlogit* command were applied to investigate the role of occupational class as a predictor of the trajectory group membership. In a two-step-analysis, estimates are given for occupational class, first, adjusted for age and first language and, second, additionally adjusted for education. In the analyses, the trajectory-group membership is treated as the outcome, where the trajectory cluster indicating the lowest healthcare utilisation was defined as the reference group and the other trajectories get the value 1 in each respective analyses. The results are given as relative risk ratios with their 95% CIs. All statistical analyses were performed with Stata 15.

## RESULTS

### Descriptive results

The study sample included 9762 the City of Helsinki employees aged between 20 and 34 at the beginning of the follow-up. Seventy-three per cent of the employees were women. Among men, the yearly average of OHS visits was 1.03 (SD, 1.44) during the mean of 1341 days of follow-up. Among women, the yearly average was 1.16 (SD, 1.50) OHS visits in the mean of 1328 days. Of the subjects of the study, 2272 (23%) were managers or professionals, 1824 (19%) semiprofessionals, 4064 (42%) routine non-manual workers and 1602 (16%) manual workers (tables 1 and 2). Managers/professionals had an equal gender distribution, semiprofessionals and routine non-manual workers were more often women, whereas men constituted the majority of manual workers.

Occupational class was closely linked to educational attainment in both genders.

In trajectory analysis, a trajectory model consisting of four distinct trajectories including one trajectory with a linear, one with a quadratic and two with a cubic shape showed the best fit using the BIC criterion (figure 1). The largest identified trajectory group 'No visits' (n=5106, 50%) represents those with less than 0.2 annual OHS visits over the 4 years of follow-up. There were two intermediate groups. The group labelled 'Low/increasing' (n=1744, 18%) is characterised by low number of visits during the first 2 years followed by a slight increase in visits during the next 2 years. The group 'Low/decreasing' (n=2238, 22%) follows a similar pattern as the group 2, but in a reverse order. The members of the both low groups averaged 1.5 annual OHS visits during the follow-up. The 'High/recurrent' group (n=976, 10%) consists of employees characterised by high levels of OHS visits, with an average of 4.6 visits per year, from the start of the employment to the end of the follow-up. The mean assignment probabilities were 0.93 for the 'No visits', 0.81 for the 'Low/increasing', 0.82 for the 'Low/decreasing' and 0.92 for the 'High/recurrent' trajectory groups, indicating a good model fit to the data. Of men, 54% belonged to the 'No visits' trajectory and 8.4% to the 'High/recurrent' trajectory group, whereas the corresponding figures for women were 50% and 10%, indicating a higher propensity for women to belong to the 'High/recurrent' trajectory group. The assignment of the members of different occupational classes to different OHS trajectories followed the socioeconomic gradient. Of managers or professionals, 59% belonged to the 'No visits' trajectory and 5% to the 'High/recurrent' trajectory. The corresponding figures for semiprofessionals were 48% and 10%, for routine non-manual workers 48% and 11% and for manual workers 49% and 13%, respectively.

### OHS utilisation trajectories by occupational class

Occupational class was a strong independent predictor for the OHS utilisation trajectories, as demonstrated in table 3. The likelihood of belonging to the 'High/recurrent' trajectory was increased for those being in a lower occupational classes when compared with the employees in managerial or professional positions. For both women and men, the risk for belonging to the 'High/recurrent' trajectory was highest for manual workers, followed by routine non-manual workers and semiprofessionals.

The independent effect of occupational class remained after adjustment for all covariates including age, first language and education. The association was most evident in the 'High/recurrent' trajectory. The relative risk for this group membership was 2.92 (95% CI 1.48 to 5.74) for male routine non-manual workers and 3.56 (95% CI 1.83 to 6.92) for male manual workers. The corresponding figures for women were 2.28 (95% CI 1.65 to 3.15) and 2.71 (95% CI 1.85 to 3.97), respectively. The results indicate that a proportion of the association between occupational class and belonging to the 'High/recurrent'

**Table 1** Descriptive statistics of the four occupational classes among 2454 male employees of the City of Helsinki aged 20–34 years

| | Total | | OHS visits p.a. | | Occupational class | | | | | | | |
| --- | --- | --- | --- | --- | --- | --- | --- | --- | --- | --- | --- | --- |
| | | | | | Managers or professionals | | Semiprofessionals | | Routine non-manual workers | | Manual workers | |
| | N | %/(SD) | N | 1/1000 | N | %/(SD) | N | %/(SD) | N | %/(SD) | N | %/(SD) |
| Total | 2454 | 100 | 2528 | 1030.3 | 594 | 100 | 286 | 100 | 677 | 100 | 897 | 100 |
| The length of the follow-up in days, average (SD) | 1341 | (252) | | | 1371 | (204) | 1382 | (191) | 1290 | (292) | 1348 | (260) |
| OHS visits per annum, 1/1000 (SD) | 1030.3 | (1435.3) | | | 705.4 | (939.4) | 996.5 | (1151.3) | 1121.5 | (1482.1) | 1187.3 | (1695.8) |
| Outcome: Trajectory group | | | | | | | | | | | | |
| 1. No OHS visits | 1337 | 54.5 | 225 | 168.5 | 374 | 63 | 148 | 51.7 | 349 | 51.6 | 466 | 52 |
| 2. Low/increasing | 420 | 17.1 | 626 | 1489.3 | 91 | 15.3 | 54 | 18.9 | 131 | 19.4 | 144 | 16.1 |
| 3. Low/decreasing | 496 | 20.2 | 728 | 1468.2 | 113 | 19 | 63 | 22 | 138 | 20.4 | 182 | 20.3 |
| 4. High/recurrent | 201 | 8.2 | 949 | 4722.6 | 16 | 2.7 | 21 | 7.3 | 59 | 8.7 | 105 | 11.7 |
| Covariates | | | | | | | | | | | | |
| Age (years) | | | | | | | | | | | | |
| 20–24 | 618 | 25.2 | 684 | 1107.2 | 23 | 3.9 | 38 | 13.3 | 221 | 32.6 | 336 | 37.5 |
| 25–29 | 1077 | 43.9 | 996 | 924.6 | 299 | 50.3 | 152 | 53.1 | 296 | 43.7 | 330 | 36.8 |
| 30–34 | 759 | 30.9 | 848 | 1117.6 | 272 | 45.8 | 96 | 33.6 | 160 | 23.6 | 231 | 25.8 |
| First language | | | | | | | | | | | | |
| Finnish | 2154 | 87.8 | 2219 | 1029.9 | 499 | 84 | 272 | 95.1 | 589 | 87 | 794 | 88.5 |
| Swedish | 81 | 3.3 | 36 | 438.3 | 47 | 7.9 | 2 | 0.7 | 24 | 3.5 | 8 | 0.9 |
| Other | 193 | 7.9 | 256 | 1327.7 | 24 | 4 | 12 | 4.2 | 62 | 9.2 | 95 | 10.6 |
| Education | | | | | | | | | | | | |
| Basic education/lower secondary | 1595 | 65 | 1807 | 1132.8 | 136 | 22.9 | 115 | 40.2 | 530 | 78.3 | 814 | 90.7 |
| Upper secondary | 476 | 19.4 | 453 | 951.2 | 126 | 21.2 | 149 | 52.1 | 130 | 19.2 | 71 | 7.9 |
| Higher education | 383 | 15.6 | 269 | 701.7 | 332 | 55.9 | 22 | 7.7 | 17 | 2.5 | 12 | 1.3 |

Results are based on register data covering the years from 2004 to 2017.
OHS, occupational health service.

**Table 2** Descriptive statistics of the four occupational classes among 7308 female employees of the City of Helsinki aged 20–34 years

| | Total | | OHS visits p.a. | | Occupational class | | | | | | | |
| | | | | | Managers or professionals | | Semi-professionals | | Routine non-manual workers | | Manual workers | |
| | N | %/(SD) | N | 1/000 | N | %/(SD) | N | %/(SD) | N | %/(SD) | N | %/(SD) |
|---|---|---|---|---|---|---|---|---|---|---|---|---|
| Total | 7308 | 100.0 | 8470 | 1158.9 | 1678 | 100.0 | 1538 | 100.0 | 3387 | 100.0 | 705 | 100.0 |
| The length of the follow-up in days, average (SD) | 1328 | (254) | | | 1326 | (244) | 1375 | (202) | 1310 | (272) | 1322 | (281) |
| OHS visits per annum, 1/1000 (SD) | 1158.9 | (1499.2) | | | 850.9 | (1114.0) | 1141.7 | (1357.4) | 1268.7 | (1627.4) | 1402.5 | (1808.0) |
| Outcome: Trajectory group | | | | | | | | | | | | |
| 1. No OHS visits | 3623 | 49.6 | 644 | 177.7 | 972 | 57.9 | 734 | 47.7 | 1599 | 47.2 | 318 | 45.1 |
| 2. Low/increasing | 1281 | 17.5 | 1922 | 1500.2 | 264 | 15.7 | 249 | 16.2 | 627 | 18.5 | 141 | 20.0 |
| 3. Low/decreasing | 1651 | 22.6 | 2455 | 1487.0 | 347 | 20.7 | 397 | 25.8 | 759 | 22.4 | 148 | 21.0 |
| 4. High/recurrent | 753 | 10.3 | 3449 | 4580.3 | 95 | 5.7 | 158 | 10.3 | 402 | 11.9 | 98 | 13.9 |
| Covariates | | | | | | | | | | | | |
| Age (years) | | | | | | | | | | | | |
| 20–24 | 2252 | 30.8 | 2693 | 1195.6 | 142 | 8.5 | 443 | 28.8 | 1412 | 41.7 | 255 | 36.2 |
| 25–29 | 3152 | 43.1 | 3565 | 1130.9 | 959 | 57.2 | 693 | 45.1 | 1233 | 36.4 | 267 | 37.9 |
| 30–34 | 1904 | 26.1 | 2212 | 1161.9 | 577 | 34.4 | 402 | 26.1 | 742 | 21.9 | 183 | 26.0 |
| First Language | | | | | | | | | | | | |
| Finnish | 6451 | 88.3 | 7627 | 1182.3 | 1403 | 83.6 | 1422 | 92.5 | 3020 | 89.2 | 606 | 86.0 |
| Swedish | 324 | 4.4 | 307 | 947.5 | 158 | 9.4 | 51 | 3.3 | 109 | 3.2 | 6 | 0.9 |
| Other | 448 | 6.1 | 479 | 1068.6 | 40 | 2.4 | 64 | 4.2 | 252 | 7.4 | 92 | 13.0 |
| Education | | | | | | | | | | | | |
| Basic education/lower secondary | 3641 | 49.8 | 4667 | 1281.8 | 273 | 16.3 | 232 | 15.1 | 2541 | 75.0 | 595 | 84.4 |
| Upper secondary | 2319 | 31.7 | 2565 | 1105.9 | 270 | 16.1 | 1211 | 78.7 | 755 | 22.3 | 83 | 11.8 |
| Higher education | 1348 | 18.4 | 1238 | 918.4 | 1135 | 67.6 | 95 | 6.2 | 91 | 2.7 | 27 | 3.8 |

Results are based on register data covering the years from 2004 to 2017.
OHS, occupational health service.

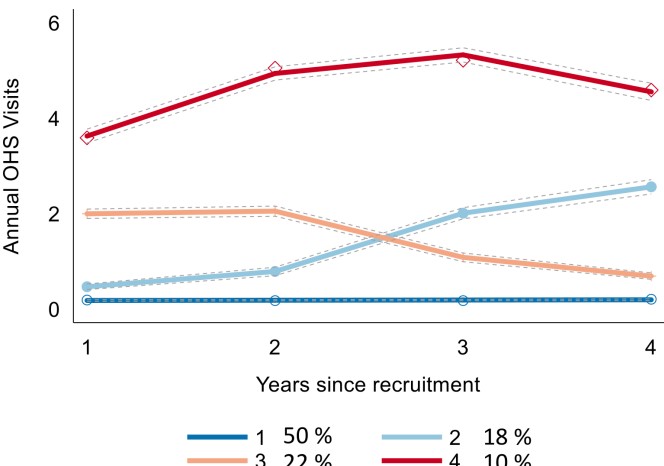

**Figure 1** Four OHS trajectories, based on registers covering the years 2004–2017 among 9762 employees of the City of Helsinki aged 20–34 years. 1=No visits, 2=Low/increasing, 3=Low/decreasing, 4=High/recurrent use. OHS, occupational health service.

trajectory is dependent on the lower educational attainment of the members of the lower occupational classes. The results comparing the two low trajectories with the 'No visits' OHS trajectory were less clear in terms of statistical significance. Whereas all estimates but one in 'Low/decreasing' versus 'No visits' comparison remained statistically significant after full adjustment, four out of 'Low/increasing' versus 'No visits' comparisons become statistically non-significant in the final model. The observed excess risks generated by occupational class were thus smaller in both 'Low' trajectories compared with 'High/recurrent' trajectory. Notably, the relative risks related to the membership of these middle trajectories were not manifested in a dose-exposure manner as was the case with the 'High/recurrent' trajectory.

### Sensitivity analyses

To assess the extent the parameter estimates were sensitive to potential errors in model specification and data, four types of sensitivity analyses were performed. First, we reproduced estimates from the original data with bootstrap resampling (1000 replications). Second, we reproduced the results with logistic regression analyses defining the high-utilisation group as those whose total number of OHS visits during the follow-up period was 10 or more (n=1484) was compared with those with no or just one visit (n=3993). Third, we ran the analysis with reversed class order in order to identify whether semiprofessionals were less likely to be in the high trajectory group compared with routine workers (see online supplementary web appendix 3). These sensitivity analyses indicated robustness of our inference about the relationship between occupational class and OHS utilisation trajectories. Fourth, we performed a sensitivity analysis where variables on part-time work and fixed-term contract were included, but they changed the estimates only modestly (data not shown).

**Table 3** Multinomial logistic regression on occupational class as a determinant of OHS trajectories among 9762 employees of the City of Helsinki aged 20–34 years

| | OHS trajectory comparison | |
| --- | --- | --- |
| | **Low/increasing versus No OHS visits** | |
| **Trajectory** | **Model 1\*** | **Model 2\*\*** |
| Men | | |
| Managers or professionals | 1.00 | 1.00 |
| Semiprofessionals | 1.55 (1.04 to 2.31) | 1.31 (0.85 to 2.03) |
| Routine non-manual workers | 1.71 (1.24 to 2.37) | 1.40 (0.96 to 2.06) |
| Manual workers | 1.38 (1.00 to 1.90) | 1.12 (0.76 to 1.66) |
| Women | | |
| Managers or professionals | 1.00 | 1.00 |
| Semiprofessionals | 1.25 (1.02 to 1.54) | 1.19 (0.92 to 1.53) |
| Routine non-manual workers | 1.47 (1.24 to 1.76) | 1.40 (1.11 to 1.76) |
| Manual workers | 1.64 (1.28 to 2.11) | 1.55 (1.16 to 2.08) |

Results are based on register data covering the years from 2004 to 2017.

| | **Low/decreasing versus No OHS visits** | |
| --- | --- | --- |
| **Trajectory** | **Model 1\*** | **Model 2\*\*** |
| Men | | |
| Managers or professionals | 1.00 | 1.00 |
| Semiprofessionals | 1.32 (0.91 to 1.91) | 1.45 (0.96 to 2.19) |
| Routine non-manual workers | 1.31 (0.97 to 1.77) | 1.53 (1.05 to 2.22) |
| Manual workers | 1.26 (0.94 to 1.68) | 1.50 (1.04 to 2.18) |
| Women | | |
| Managers or professionals | 1.00 | 1.00 |
| Semiprofessionals | 1.48 (1.24 to 1.77) | 1.71 (1.36 to 2.15) |
| Routine non-manual workers | 1.33 (1.13 to 1.56) | 1.67 (1.34 to 2.07) |
| Manual workers | 1.30 (1.02 to1.65) | 1.65 (1.25 to 2.19) |

| | **High/recurrent versus No OHS visits** | |
| --- | --- | --- |
| **Trajectory** | **Model 1\*** | **Model 2\*\*** |
| Men | | |
| Managers or professionals | 1.00 | 1.00 |
| Semiprofessionals | 3.21 (1.62 to 6.36) | 2.60 (1.23 to 5.49) |
| Routine non-manual workers | 3.99 (2.22 to 7.17) | 2.92 (1.48 to 5.74) |
| Manual workers | 5.02 (2.86 to 8.80) | 3.56 (1.83 to 6.92) |
| Women | | |
| Managers or professionals | 1.00 | 1.00 |
| Semiprofessionals | 2.13 (1.61 to 2.81) | 2.29 (1.62 to 3.24) |

Continued

**Table 3** Continued

| Trajectory | High/recurrent versus No OHS visits | |
| --- | --- | --- |
| | Model 1* | Model 2** |
| Routine non-manual workers | 2.53 (1.98 to 3.25) | 2.28 (1.65 to 3.15) |
| Manual workers | 3.10 (2.25 to 4.27) | 2.71 (1.85 to 3.97) |

Model 1*=Model adjusted for age and first language, Model 2**=M1+education.
OHS, occupational health service.

## DISCUSSION

In this study, we identified developmental trajectories and socioeconomic differences in OHS primary care service utilisation among 20–34-year-old employees of the City of Helsinki from 2004 to 2017. Our key results were: (1) Half of the young employees did not use OHS to any considerable extent. (2) Higher occupational classes used less OHS. (3) Four trajectory groups, that is, 'No visits', 'Low/increasing', 'Low/decreasing' and 'High/recurrent', were identified. (4) The trajectory group of 'High/recurrent' included a larger number of lower class workers, especially among men, and the differences were large also among women. (5) Occupational class differences in 'Low/decreasing' group were evident in both genders. (6) Only in women there were some occupational class differences in belonging to the trajectory group of 'Low-increasing OHS utilisation'.

Our results highlight the significance of socioeconomic gradient in OHS utilisation that was visible both in men and women. The percentage of those who had no visits was the highest among managers and professionals and the proportion of no visits decreased when going down the occupational class ladder. Respectively, high and recurrent use was smallest among managers and professionals and increased with decreasing occupational class, this type of use being the most common among manual workers. A larger proportion of men had no visits at all in each occupational class, thus the women used the health services more, in line with earlier findings.[2] In a similar way, the high and recurrent use was higher among women than among men. In the present context, primary care visits can be interpreted as an indicator of incidence of acute illnesses, as the Finnish OHS system distinguish visits related to occupational health hazards. The present results concerning primary care visits are in line with previous findings from our own and other studies showing the socioeconomic differences in SA among employees and the gender differences in SA, that is, women having more absence than men.[7 10 14 18 21] It can be assumed that large number of OHS visits precede SA.[27]

Stratified analyses indicated gender differences in OHS utilisation. According to our results, among men, the occupational class differences disappeared after full adjustment in the trajectory group of 'Low/increasing'. This implies that the initial differences are associated with the type of work tasks. In contrast, after full adjustment,

among women the employees in the two lowest occupational classes had a higher risk for belonging to this trajectory group. In the trajectory group of 'Low/decreasing', the differences were initially similar among the three lowest occupational classes. However, the differences modestly increased after full adjustment, implying that there are several factors associated with the low OHS utilisation. This was seen among both genders.

The trajectory group of 'High/recurrent' is perhaps the most interesting group alongside with the 'Low/increasing' group in terms of costs and possible preventive opportunities. According to our results the occupational class differences in this group are steep especially among men and also large among women. After full adjustment, the differences decreased more in men, suggesting that the initial differences are more associated with work tasks among them. However, the differences remained high in both genders after adjustments. Studies regarding socioeconomic differences in SA have also found that the differences are steeper among men,[7 17 18] but the former studies mostly concentrate on older employees.

The differences in physical and psychosocial demands between occupational classes are important to take into account when interpreting the results. Manual workers have more physically demanding jobs, which may affect their need for primary care services. Adverse working conditions may cause ill-health and need of health services as milder even health difficulties may prevent these employees from working. Employees in higher occupational classes typically have more complex and mentally demanding jobs.[28] In studies examining the socioeconomic differences in SA, physical working conditions have been found to be the strongest explanatory factor.[12 16] However, employees in our study are fairly young and thus adverse physical or psychosocial working conditions might not have yet affected their health, as health-related effects usually increase with age. In addition, the unique OHS system where the visits associated with occupational health hazards (preventive services) are recorded separately from primary care visits may contribute to the differences seen in our results. For example, visits with more chronic work-related reason are usually not recorded as primary care visits. Thus, the overall utilisation of OHS requires further research.

Our study indicates that service use patterns might recognise vulnerable groups more precisely than just belonging to certain occupation or occupational class may do. Despite this, the two lowest occupational classes may need extra attention based on their OHS utilisation patterns. Case management protocols are essential in coordinating patient-centred care path which also saves costs.[8 9 29] Among younger employees, timely treatment is highly important, as it might prevent the worsening of their condition. OHS should identify those employees who use services a lot.

Previous longitudinal studies using OHS data are lacking, but recent studies have showed that frequent utilisation of OHS was associated with psychiatric problems

and musculoskeletal disorders,[2] whereas the latter also predicts persistent frequent utilisation.[30] Furthermore, frequent utilisation has increased the risk of long SA[31] and disability pension.[32] These associations highlight the need of identifying those in risk for more severe illness and work disability at an early stage, and information on the different utilisation trajectories with identified occupational class differences supports these preventive actions. Moreover, some comparisons can be made with the studies investigating frequent attenders in primary care. Frequent attenders in primary care in the general population have been studied particularly in Netherlands and in Sweden using questionnaire surveys and record linkage. These studies had participants from a wide range of sociodemographic backgrounds and they consider only visits to general practitioners, thus their direct comparability to OHS utilisation is difficult. In addition, the definition of frequent attender varies between studies.[33] However, previous studies have found out that frequent attenders have multiple reasons for presenting,[34] but overall chronic illnesses,[33] somatic diseases and symptoms[35 36] and especially psychiatric problems[35] have been associated with more frequent primary care service use. Frequent attenders have more health discomfort, low mastery and they may be more vulnerable for stressful life events due to inadequate coping strategies.[27 37 38] In line with the study by Reho et al,[31 32] frequent attenders are a high-risk group for long-term SA and disability pension.[27] According to two previous Dutch studies, one out of every seven 1-year frequent attenders becomes persistent frequent attender and six out of seven are transient frequent attenders.[35 39] Based on this previous evidence, the inclusion of diagnostic information would be important in future studies of OHS utilisation. However, from the methodological viewpoint both the Dutch and Swedish research groups point out that age should be taken into account when studying the frequent attenders, as the reasons for high service utilisation and what constitutes high use highly differ by age group.[36 40]

## Methodological considerations

The registers used in this study are reliable and comprehensive. We focused on all occupational groups within the largest employer in Finland and the sample consisted all of 20–34-year-old employees within this organisation. The occupational healthcare policies are same for all these employees and have remained same during the study period of 2004–2017. Our study avoids a common limitation of OHS studies that are based on limited samples with data on only those attending the service (eg, Reho et al[2]) or those responding to survey.[3] Another advantage of this study is that we could make inferences based on longitudinal cohort data instead of relying on cross-sectional evidence. Limitations of the data include the lack of information of diagnoses, physical and psychosocial working conditions, lifestyle factors and any primary care visits outside the OHS. Unfortunately, our data do not extend beyond the employees' current employment

contracts. In addition, people with initial poor health may attain lower educational level and end up in lower occupational positions. Moreover, excluding those with employment record for less than 4 years (n=12 512) due to the need of long enough follow-up time reduced particularly the number of youngest employees (see online supplementary web appendices 1 and 2). A further limitation is that the initial occupational classes might have changed during the follow-up for due to promotion or other changes in the employment.

The OHS system in Finland is unique, thus comparison to other countries is difficult. The principle of primary care use is similar as in the general practice or family doctor setting in most other Western countries, but the patient population differs to some extent from ours in terms of demographics and employment status. In Finland, the employer offers (most employers do) those services and thus OHS is the main source for primary care for employees due to being free at the point of delivery and enabling an easy access. However, even within Finland, different employers can have different policies in terms of provision of primary care services. Nevertheless, our results can be broadly generalised to Finnish public sector employees.

The present study is to our knowledge the first one that used longitudinal latent class analysis aiming to capture OHS utilisation as a complex longitudinal phenomenon. GBTM approach mixes the application of formal statistical criteria and subjective evaluation in model fitting.[24] One of its strength is that it allowed us to identify high OHS utilisation over time. It is a limitation that those who left the City of Helsinki within the first 4 years of their employment were lost to follow-up. Another benefit is that GBTM is capable of identifying different OHS trajectories within subjects that appear similar in terms of summary statistics. In this study, we were able to distinguish between two 'low' trajectories, which may allow for better planning of targeted prevention measures. However, we want to highlight that by this methodology it cannot be ascertained that the observed subgroups are distinct population subgroups. As in case of any latent trajectory class analyses, there is a possibility that the data could be interpreted as homogenous but non-normal.[41] We find, however, the obtained groups to be realistic and the results applicable in terms of real-life interpretations. However, further analysis with longer follow-up would be important to confirm the trajectories found.

## CONCLUSION

We used GBTM for distinguishing four different developmental trajectories in OHS primary care service utilisation among 20–34-year-old employees of the City of Helsinki. We found that occupational class differences exist in the utilisation development trajectories. A large proportion of the young employees do not use OHS primary care services and non-use is the most common among the highest occupational class, especially trajectories where

the utilisation has grown or been high throughout the follow-up had large occupational class differences, which followed the socioeconomic gradient. Identifying high utilisation patterns is important as 10% of employees that may be labelled high and recurrent users account for 40% of the all OHS consultations.

Our results show several important points for policy makers as well as occupational and primary healthcare personnel in Finland and in countries with different primary care and occupational healthcare systems. According to our results, preventive measures should be considered particularly among the trajectory groups of 'Low/increasing' and 'High/recurrent' healthcare utilisation. In addition, special attention should be paid to the lowest occupational classes, and their OHS utilisation should be closely monitored by the occupational healthcare/primary care in order to identify those in need for extra support. Case management protocols should be further developed and resources targeted in order to develop and maintain the healthcare system where early support is been given to those identified being in risk for subsequent work disability. As the preventive measures are done in practice, research should follow their success and produce evidence based development suggestions. In addition, OHS and primary care utilisation requires more longitudinal research in order to target resources and preventive measures.

**Contributors** HS, AK and JH designed the study. HS mainly wrote the manuscript with contributions and comments from all the other authors (JH, KP, OP, OR and AK). JH did the statistical analyses and wrote the results section and OP commented those. All the authors have approved the final version of the manuscript.

**Funding** This study was funded by the Finnish Work Environment Fund (grant 117321) and the Academy of Finland (grant 315343). AK was funded by the Economic and Social Research Council (ESRC) (grant ES/L007509/1 and ES/S00744X/1). OR was funded by the Academy of Finland (grant 1294514).

**Disclaimer** The funders had no involvement to the preparation of the manuscript.

**Competing interests** None declared.

**Patient consent for publication** Not required.

**Provenance and peer review** Not commissioned; externally peer reviewed.

**Data availability statement** No data are available.

**ORCID iDs**
Hilla Sumanen http://orcid.org/0000-0001-9641-6518
Jaakko Harkko http://orcid.org/0000-0001-8682-1544
Anne Kouvonen http://orcid.org/0000-0001-6997-8312

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
