## [Reviewer comments · BMJ Open]

ARTICLE DETAILS

TITLE (PROVISIONAL)	The association between socio-economic position and occupational health service utilization trajectories among young municipal employees in Finland
AUTHORS	Sumanen, Hilla; Harkko, Jaakko; Piha, Kustaa; Pietilainen, Olli; Rahkonen, Ossi; Kouvonen, Anne

VERSION 1 - REVIEW

REVIEWER	Lu Liming Clinical Research Center, South China Research Center for Acupuncture and Moxibustion, Medical College of Acu-Moxi and Rehabilitation, Guangzhou University of Chinese Medicine, China
REVIEW RETURNED	19-Mar-2019

GENERAL COMMENTS	1. In the abstract, follow-up data for four years were included in the present study. Could you explain how the factors adverse physical , psychosocial working conditions or occupational stress affect the utilization trajectories of primary care services provided by OHS in 4 years? 2. In the last sentence of Introduction, the hypothesis that lower occupational classes have a higher risk of high OHS utilization, when differences in terms of personal characteristics (age, education, first language) are brought into the analysis. Could you give more details to explain how to define the low or high of occupational classes. 3. Please state the method details in the main text. What is the basis for inclusion and exclusion criteria? (Page 5) 4. In the method, occupational class was measured with four classes: "managers and professionals" (e.g. teachers and physicians), "semi-professionals" (e.g. nurses and foremen), "routine non-manuals" (e.g. clerical employees and childminders) and "manual workers" (e.g. technical and cleaning staff). Is there a basis for this classification of professional classes? Could you provide some citation? 5. Different groups of graphs do not match the legends in Figure 1. Please confirm it (Page 26). 6. In the discussion, it was said that the OHS system in Finland is unique, thus comparisons to other countries is difficult. Whether the conclusions of this study offered a certain reference for for
---

	OHS systems in other regions or countries. Please give more details. 7.How did you deal with the data of employees who were lost to follow-up in analysis.
--	--

REVIEWER	Linn Holness University of Toronto Canada
REVIEW RETURNED	29-Mar-2019

GENERAL COMMENTS	This is an interesting article that presents useful information and is valuable use of available data. While it is relatively unique to Finland, it is nonetheless worthwhile. Maybe the authors could comment if there are other jurisdictions that have similar systems to allow the reader to understand how unique it is. Given that the two time periods are just 2 years, I wonder if an alternate explanation for the for low increasing versus low decreasing groups is because of an acute event that led to increased use of the system but would not represent a true trend if the follow-up period was longer.
---

REVIEWER	Peter Tammes University of Bristol (United Kingdom)
REVIEW RETURNED	29-Mar-2019

GENERAL COMMENTS	As the study states to make better use of the Helsinki occupational health services (OHS), it's aims are unclear. As described in the introduction, it will identify developmental trajectories, test a hypothesis, and predict outcomes. I think the study's first part is descriptive or exploratory when identifying trajectories, and the second part is associating trajectories with demographic and socio-economic characteristics. This is rather explaining then predicting. The stated hypothesis is not clearly introduced and seems not to fit in this study. The study includes employees aged between 20 and 34 years of age. It is not clear why the authors have chosen this age range; is there a medical/clinical reason or is it that young adults' transitions to independence in Western world have become increasingly protracted and non-linear. The study uses longitudinal data from the Helsinki OHS. However, it doesn't conduct a longitudinal analysis as it counts annual numbers of visits. Following individual employees 4 years, allows to examine changes in socio-economic characteristics over time with differences in OHS use. This can be done when applying a multi-level model whereby time is level-1 unit and employee level-2 unit. Within this study, it is assumed, for example, that no-ones makes promotion within the observation time. If I am correct, the age at the start of the person's observation period is taken here. The categorisation of age might be problematic as healthcare service use is age related. The totals given in the table 1 titles, for both men and women, do not match with the total N given in the table, for example '2,678 male' whereas the N is 2454. What explains the differences; the tables present only complete cases? Is there any information on
--

	OHS use of employees whose employment record was less than 4 years (left the job). Table 1 shows nearly two third of the managers/professionals had no OHS visit over 4 years. The other groups show substantial lower percentages. Is it possible that this occupational group makes more often use of other (e.g. private) healthcare services? As the data are rather used in a cross-sectional way, another option might be reverse causation: only healthy employees (i.e. don't use recorded OHS) make job promotion or are appointed in higher functions. The study adjusted for age, first language, and education. Are there any other potential confounders the authors could include in their model, such as part-time work, or otherwise to be discussed in the Discussion section? The comparison with studies focusing on frequent attenders in primary care is interesting. Within the UK studies also have examined frequent attenders at primary care and, particularly, secondary care (Accident & Emergency departments).
--	--

REVIEWER	Ayumi Shintani, PhD, MPH Osaka City University, Osaka Japan
REVIEW RETURNED	09-Jun-2019

GENERAL COMMENTS	Statistical analyses were well presented and performed using an advanced method, GBTA. The analysis included only subjects who had a full 4 years of data, the results could be biased due to missing data. On page 14 (lines 53-56), the authors stated as missing data could be random by saying "However, we want to highlight that by this methodology it cannot be ascertained that the observed subgroups are distinct population subgroups", there is no clear explanation to endorse this sentence. The authors at least should perform analyses to compare background between those included and excluded in the analysis.
--

VERSION 1 – AUTHOR RESPONSE

Reviewer(s)' Comments to Author:

Reviewer: 1

Reviewer Name: Lu Liming

Institution and Country: Clinical Research Center, South China Research Center for Acupuncture and Moxibustion, Medical College of Acu-Moxi and Rehabilitation, Guangzhou University of Chinese Medicine, China

1. In the abstract, follow-up data for four years were included in the present study. Could you explain how the factors adverse physical, psychosocial working conditions or occupational stress affect the utilization trajectories of primary care services provided by OHS in 4 years?

- This is a register-based study and therefore working conditions and occupational stress were not included in the data. We have clarified this and added it as a limitation (page 15):

“Limitations of the data include the lack of information of diagnoses, physical and psychosocial working conditions, lifestyle factors and any primary care visits outside the OHS.”

2. In the last sentence of Introduction, the hypothesis that lower occupational classes have a higher risk of high OHS utilization, when differences in terms of personal characteristics (age, education, first language) are brought into the analysis. Could you give more details to explain how to define the low or high of occupational classes.

- We have added more details about the definitions of occupational classes to the Methods section, pages 6-7:

“We used four occupational class groups. Based on the socioeconomic classification of Statistics Finland and the occupational classification of the City of Helsinki (Laaksonen et al. 2010, Lahelma et al. 2013), non-manual employees were divided into three groups based on skills requirements and supervisory status: Managers and professionals, semi-professionals, and routine non-manual employees. Managers have subordinates and they do managerial or administrative work, whereas professionals include employees with a university degree, such as physicians and teachers. Semi-professionals include occupations such as registered nurses and technicians. Routine non-manual employees include clerical employees and lesser-educated occupations particularly within the social and health care, such as child-minders and care workers. The fourth occupational class, manual workers, include occupations for example from the fields of cleaning, kitchen work and public transport.”

3. Please state the method details in the main text. What is the basis for inclusion and exclusion criteria? (Page 5)

- We have clarified the inclusion and exclusion criteria in the beginning of Methods section, page 6 as follows:

“We excluded employees with less than four years of employment history as we needed a long enough follow-up time to observe potential development trajectories.”

4. In the method, occupational class was measured with four classes: “managers and professionals” (e.g. teachers and physicians), “semi-professionals” (e.g. nurses and foremen), “routine non-manuals” (e.g. clerical employees and childminders) and “manual workers” (e.g. technical and cleaning staff). Is there a basis for this classification of professional classes? Could you provide some citation?

- Please see above response 2. We have clarified this and provided more information.

5. Different groups of graphs do not match the legends in Figure 1. Please confirm it (Page 26).

- Thank you for pointing that out, we have revised the legends.

6. In the discussion, it was said that the OHS system in Finland is unique, thus comparisons to other countries is difficult. Whether the conclusions of this study offered a certain reference for OHS systems in other regions or countries. Please give more details.

- Thank you for this important comment. We have added clarification to Introduction and Discussion:

Introduction (page 4):

“Finland has a unique occupational health service (OHS) system with statutory prevention of occupational health hazards (preventive services) and additionally purchased primary care services. OHS may be provided by employer’s own OHS units, private clinics or public sector health centers with specifically trained occupational health physicians, nurses, physiotherapists and psychologists. In Finland, most employers purchase the additional primary care element for their employees. In 2017, 94% of the employees covered by statutory preventive OHS also had an access to primary care for any illness including all non-work related illnesses, paid by their employer and partly subsidized by the National Pension Fund [1]. OHSs are free for employees at the point of delivery and their accessibility is typically good, making them the main source and the preferred type of primary care for Finnish employees. “

Discussion (pages 15-16):

“The OHS system in Finland is unique, thus comparison to other countries is difficult. The principle of primary care use is similar as in the general practice (GP) or family doctor setting in most other Western countries, but the patient population differs to some extent from ours in terms of demographics and employment status. In Finland, the employer offers (most employers do) those services and thus OHS is the main source for primary care for employees due to being free at the point of delivery and enabling an easy access. However, even within Finland, different employers can have different policies in terms of provision of primary care services. Nevertheless, our results can be broadly generalized to Finnish public sector employees.”

7.How did you deal with the data of employees who were lost to follow-up in analysis.

- We included only those employees who were employed by the City of Helsinki for at least 4 years. For this reason there is technically no loss to follow-up but, instead, relatively high number of excluded subjects. We have produced web-appendixes 1 and 2 that present descriptive statistics for included and excluded subjects.

Reviewer: 2

Reviewer Name: Linn Holness

Institution and Country: University of Toronto - Canada

This is an interesting article that presents useful information and is valuable use of available data. While it is relatively unique to Finland, it is nonetheless worthwhile.

- Thank you for these positive comments.

Maybe the authors could comment if there are other jurisdictions that have similar systems to allow the reader to understand how unique it is.

- Thank you for this important comment. We have added clarification to Introduction and Discussion:

Introduction (page 4):

“Finland has a unique occupational health service (OHS) system with statutory prevention of occupational health hazards (preventive services) and additionally purchased primary care services. OHS may be provided by employer’s own OHS units, private clinics or public sector health centers with specifically trained occupational health physicians, nurses, physiotherapists and psychologists. In Finland, most employers purchase the additional primary care element for their employees. In 2017, 94% of the employees covered by statutory preventive OHS also had an access to primary care for any illness including all non-work related illnesses, paid by their employer and partly subsidized by the National Pension Fund [1]. OHSs are free for employees at the point of delivery and their accessibility is typically good, making them the main source and the preferred type of primary care for Finnish employees. “

Discussion (pages 15-16):

“The OHS system in Finland is unique, thus comparison to other countries is difficult. The principle of primary care use is similar as in the general practice (GP) or family doctor setting in most other Western countries, but the patient population differs to some extent from ours in terms of demographics and employment status. In Finland, the employer offers (most employers do) those services and thus OHS is the main source for primary care for employees due to being free at the point of delivery and enabling an easy access. However, even within Finland, different employers can have different policies in terms of provision of primary care services. Nevertheless, our results can be broadly generalized to Finnish public sector employees.”

Given that the two time periods are just 2 years, I wonder if an alternate explanation for the for low increasing versus low decreasing groups is because of an acute event that led to increased use of the system but would not represent a true trend if the follow-up period was longer.

- This is an important point. The follow-up time was 4 years and it is true that the trajectories identified have OHS utilization changes during 2 years. Later on, in another study, we hope that we are able to extend the follow-up time and investigate if the similar utilization trajectories still apply. We have added this limitation to Discussion, where we now write as follows: (page 16).

“We find, however, the obtained groups to be realistic and the results applicable in terms of real-life interpretations. However, further analysis with longer follow-up would be important to confirm the trajectories found.”

Reviewer: 3

Reviewer Name: Peter Tammes

Institution and Country: University of Bristol (United Kingdom)

As the study states to make better use of the Helsinki occupational health services (OHS), it’s aims are unclear. As described in the introduction, it will identify developmental trajectories, test a hypothesis, and predict outcomes. I think the study’s first part is descriptive or exploratory when identifying trajectories, and the second part is associating trajectories with demographic and socio-economic characteristics. This is rather explaining then predicting. The stated hypothesis is not clearly introduced and seems not to fit in this study.

- Thank you for these comments. We have now revised the aims and hypotheses to make them clearer, page 5:

“In our study, we aimed to identify developmental trajectories of OHS primary care service utilization among 20-34-year-old municipal employees of the City of Helsinki. In the second stage of the analysis we aimed to identify occupational class differences in belonging to different trajectory groups. We tested two hypothesis, first, that the distinct trajectories can be identified, and second, that occupational class gradient can be found in OHS utilization.”

The study includes employees aged between 20 and 34 years of age. It is not clear why the authors have chosen this age range; is there a medical/clinical reason or is it that young adults' transitions to independence in Western world have become increasingly protracted and non-linear.

- The occurrence of illness and the patterns of health care use are different in different age groups. We have previously examined the incidence of sickness absence among young municipal employees and used this age group in those studies (for example Sumanen 2016, 2017a, 2017b). Before conducting those studies, we tested many different age group options in order to find the most optimal range. Under 35-year-olds were also selected because, as suggested, the transitions to independence in Finland are increasingly protracted especially in terms of long-time/permanent employment. Therefore decreasing the upper age limit would have significantly reduced our sample size. In addition, the European Commission Eurostat defines young employees as those aged 15 to 34 (Eurostat, 2019). In the present study, we excluded those under the age of 20, because at that age it is very rare to be employed on a long-term contract in the public sector.

We added to the text, page 6:

“The selection of this age group was based on the Eurostat definition of young employees ,and on previous studies investigating the occurrence of illness and sickness absence in employees of different ages (22, 23).“

(Sumanen H. Work disability among young employees: Changes over time and socioeconomic differences. *Dissertationes Scholae Doctoralis Ad Sanitatem Investigandam Universitatis Helsinkiensis*; nro 18/2016.

Sumanen H, Lahelma E, Pietiläinen O, Rahkonen O. The Magnitude of Occupational Class Differences in Sickness Absence: 15-Year Trends among Young and Middle-Aged Municipal Employees. *International Journal of Environment Research and Public Health*, 2017a, 14, E625.

Sumanen H, Pietiläinen O, Mänty M. Self-certified sickness absence among young municipal employees – changes from 2002 to 2016 and occupational class differences. *International Journal of Environment Research and Public Health*, 2017b, 14, 1131.

Eurostat. 2019. Participation of Young People in Education and the Labour Market.

https://ec.europa.eu/eurostat/statistics-explained/index.php?title=Participation_of_young_people_in_education_and_the_labour_market&oldid=433402#Participation_of_young_persons_in_formal_education_and_in_the_labour_market. (Accessed 6 Aug 2019))

The study uses longitudinal data from the Helsinki OHS. However, it doesn't conduct a longitudinal analysis as it counts annual numbers of visits. Following individual employees 4 years, allows to examine changes in socio-economic characteristics over time with differences in OHS use. This can

be done when applying a multi-level model whereby time is level-1 unit and employee level-2 unit. Within this study, it is assumed, for example, that no-ones makes promotion within the observation time. If I am correct, the age at the start of the person's observation period is taken here. The categorisation of age might be problematic as healthcare service use is age related.

- Thank you for pointing this out. Group-based trajectory modeling (GBTM) is a methodology for approximation of an unknown and possibly complex data distribution in longitudinal data. In this method, certain phenomenon is observed through subsequent time intervals over a certain time period. GBTM is longitudinal in that each consecutive observation is assumed to be dependent on preceding observations. However, it is true that the analysis does not fully capture the possibilities of the longitudinal nature of the data. The selected analytical strategy where the trajectories are first identified on their own merit and then examining the associations between independent variables and observed trajectories is widely used, and excluding time-varying effects of the covariates from the analyses was assessed to maintain a balance between the inherent complexity of longitudinal analyses and the analytical clarity.

The totals given in the table 1 titles, for both men and women, do not match with the total N given in the table, for example '2,678 male' whereas the N is 2454. What explains the differences; the tables present only complete cases? Is there any information on OHS use of employees whose employment record was less than 4 years (left the job).

- Thank you for pointing that out, it was an error. We have revised the titles.

- We have information on OHS also of those employees whose employment record was less than 4 years. If requested we can provide some descriptive data of that group.

Table 1 shows nearly two third of the managers/professionals had no OHS visit over 4 years. The other groups show substantial lower percentages. Is it possible that this occupational group makes more often use of other (e.g. private) healthcare services?

- The results are in line with substantial knowledge of health differences; the highest occupational class group having the highest percentage of "no visits" implies that they have been healthy and not needed primary care services. Of course there is a possibility that they have used private services or municipal health care center, but in Finland the OHS is the main source of primary care service for those who are employed, as the services are free (paid by the employer) and easy-access.

As the data are rather used in a cross-sectional way, another option might be reverse causation: only healthy employees (i.e. don't use recorded OHS) make job promotion or are appointed in higher functions.

- This is an important remark. This option is also in line with the knowledge of health differences, causation and selection. We have now acknowledged the issue of reverse causation in Discussion on page 16: "In addition, people with initial poor health may attain lower educational level and end up in lower occupational positions. Unfortunately, our data does not extend beyond the current employment relations. "

We consider the effect of reverse causation related to job promotion to be limited, as formal qualifications are typically required for positions in the Finnish public sector, and changes in occupational position are rare in the four-year time period.

The study adjusted for age, first language, and education. Are there any other potential confounders the authors could include in their model, such as part-time work, or otherwise to be discussed in the Discussion section?

- When designing the study we tested several cofounders and selected those, which are based on the previous literature the main confounders in this subject. In particular, part-time work (less than 32 hours a week) and fixed-term contract are two confounders that could have been but are not presented in this study. The main reason is, to our judgement, that the disadvantage of increased complexity of the presentation would have out-weighted the gains from presenting relatively modest changes in the estimates. We added a notion on this matter on page 12: "Third, we performed a sensitivity analysis where variables on part-time work and fixed-term contract were included, but they changed the estimates only modestly." For example, a "model 3" part-time work and fixed-term contract were included would lower the estimate for being in "High recurrent trajectory" from 3.56 (95% CI, 1.83 – 6.92) to 3.24 (95% CI, 1.03 – 5.24) for male manual workers, 2.92 (95% CI, 1.48 – 5.74) to 2.81 (95% CI, 1.33 – 5.96) for male routine non-manuals and 2.60 (95% CI, 1.23 – 5.49) to 2.32 (95% CI, 1.03 – 5.24) for male semi-professionals. The effect of part-time work and fixed-term contract would be interesting to examine in the further studies, and even more so with time-varying effects. However, these variables were omitted in this study to achieve more balanced presentation of an already complex topic.

The comparison with studies focusing on frequent attenders in primary care is interesting. Within the UK studies also have examined frequent attenders at primary care and, particularly, secondary care (Accident & Emergency departments).

- Thank you for this comment, we have become familiar with the UK studies. In particular, we have read with a great interest about the findings regarding patient satisfaction towards primary care provision and primary care being easy access, and how their associations with the use of secondary care. We will keep them in mind when conducting further studies which could utilize also hospital visits. We have added references from Finland regarding occupational health service primary care utilization, which have been published during the review process, page 14:

"Previous longitudinal studies using OHS data are lacking, but recent studies have showed that frequent utilization of OHS was associated with psychiatric problems and musculoskeletal disorders [2], whereas the latter also predicts persistent frequent utilization [30]. Furthermore, frequent utilization has increased the risk of long sickness absence [31] and disability pension [32]. These associations highlight the need of identifying those in risk for more severe illness and work disability at an early stage, and information on the different utilization trajectories with identified occupational class differences supports these preventive actions."

(2. Reho T, Atkins S, Talola N, Viljamaa M, Sumanen M, Uitti J. Frequent attenders in occupational health primary care: A cross-sectional study. *Scand J Public Health* 2018;1:1403494818777436.

30. Reho T, Atkins S, Talola N, Sumanen M, Viljamaa M, Uitti J. Comparing occasional and persistent frequent attenders in occupational health primary care - a longitudinal study. *BMC Public Health* 2018;26:1291.

31. Reho TTM, Atkins SA, Talola N, Sumanen MPT, Viljamaa M, Uitti J. Occasional and persistent frequent attenders and sickness absences in occupational health primary care: a longitudinal study in Finland. *BMJ Open* 2019;19:e024980.

32. Reho TTM, Atkins SA, Talola N, Sumanen MPT, Viljamaa M, Uitti J. Frequent attenders at risk of disability pension: a longitudinal study combining routine and register data. Scand J Public Health 2019;11:1403494819838663.)

Reviewer: 4

Reviewer Name: Ayumi Shintani, PhD, MPH

Institution and Country: Osaka City University, Osaka Japan

Please state any competing interests or state 'None declared': None

Please leave your comments for the authors below

Statistical analyses were well presented and performed using an advanced method, GBTA. The analysis included only subjects who had a full 4 years of data, the results could be biased due to missing data. On page 14 (lines 53-56), the authors stated as missing data could be random by saying "However, we want to highlight that by this methodology it cannot be ascertained that the observed subgroups are distinct population subgroups", there is no clear explanation to endorse this sentence. The authors at least should perform analyses to compare background between those included and excluded in the analysis.

- Thank you for pointing this out. First we want to make a terminological clarification. Typically, in longitudinal studies subjects with missing data after inclusion are considered to be lost to follow-up. In this study subjects with less than four years of data were excluded from the study population. For the included subjects, there was practically no loss to follow-up. We have provided a web-appendixes 1 and 2, where descriptive statistics are provided both included and excluded population groups.

The quoted passage in the comment does not refer to the missing data but, rather, to the fact that the presented groupings are expected to represent an approximation of a previously unknown longitudinal (complex) data distribution rather than distinct, i.e. "true", sub-populations.

VERSION 2 – REVIEW

REVIEWER	Lu Liming Guangzhou University of Chinese Medicine
REVIEW RETURNED	15-Aug-2019

GENERAL COMMENTS	Good revision, suggest accepted and published.
--

REVIEWER	Peter Tammes University of Bristol (United Kingdom)
REVIEW RETURNED	10-Sep-2019

GENERAL COMMENTS	I have reviewed an earlier version of the manuscript. Some parts of the revised manuscript are substantially re-written including the title. However, I still have some comments and concerns.
--

	I do think occupational class can be associated with health service use based on the data, but don't think it can 'explain' as the title suggests. As the study period is 2004 till 2017, I wonder whether within that time-frame changes have been introduced in the health care system which may impact health care service utilization. Including only employees with a record of four or more years might introduce a bias as those who have left the job earlier might be a specific group and this selection might particularly impact certain age groups or occupational classes. Should employees have been employed 4 years or more consecutively or could there be gaps or breaks in their employment contract (are those who returned in- or excluded?). It is unclear when age is measured; is that at the beginning of the follow-up period of an employee? I am not sure whether categorisation of age is appropriate as, for example, someone aged 24 will only be for the first year of the follow up period in the first category and for the other three years in the second age category. Also, when following employees for 4 years or more one might expect promotions or switches between occupational class groups, e.g. semi-prof and managers? All these points should be discussed in the Method and/or Discussion section. Within the Result section the mean days of follow-up is 1341 for men and 1328 for women. If employees are followed for 4 years or more one might expect a higher mean. It is stated in the Result section that belonging to the high/recurrent trajectory was highest among manual workers, followed by routine and non-manual workers and semi-professionals. The text should include that this is compared to managers and no OHS visits. Furthermore, the presented 95% confidence intervals don't suggest such a ranking. To present such as ranking, the authors should change the reference group to see, for example, whether semi-professionals (new reference group) are less likely in the high trajectory compared to routine workers. Besides, of more interest might be which occupational class is more likely to be in the high/decrease vs high/recurrent trajectory, and similarly no OHS visits vs low/increase trajectory. Within the discussion it is expressed that this system is unique for Finland, Still, the authors should do their best to discuss and highlight what academics, doctors, and policy makers outside Finland could learn from the study.
--	--

VERSION 2 – AUTHOR RESPONSE

Reviewer(s)' Comments to Author:

Reviewer: 1

Reviewer Name: Lu Liming

Institution and Country: Guangzhou University of Chinese Medicine

Please state any competing interests or state 'None declared': None declared

Please leave your comments for the authors below

Good revision, suggest accepted and published.

- Thank you.

Reviewer: 3

Reviewer Name: Peter Tammes

Institution and Country: University of Bristol (United Kingdom)

Please state any competing interests or state 'None declared': None declared

Please leave your comments for the authors below

I have reviewed an earlier version of the manuscript. Some parts of the revised manuscript are substantially re-written including the title. However, I still have some comments and concerns.

I do think occupational class can be associated with health service use based on the data, but don't think it can 'explain' as the title suggests.

- We have revised the title based on this comment and editorial wishes: "The association between socio-economic position and occupational health service utilization trajectories among young municipal employees in Finland".

As the study period is 2004 till 2017, I wonder whether within that time-frame changes have been introduced in the health care system which may impact health care service utilization.

- We have now clarified this in the Methods section and this is also pointed out in the methodological considerations and in the "Strengths and limitations of this study" -bullet points: The OHS primary care services offered to the employees have remained same during the whole study period. (page 6)

In addition, the Finnish health care system outside of occupational health care system has not changed during our study period, however, this has not been mentioned in the text.

Including only employees with a record of four or more years might introduce a bias as those who have left the job earlier might be a specific group and this selection might particularly impact certain age groups or occupational classes. Should employees have been employed 4 years or more consecutively or could there be gaps or breaks in their employment contract (are those who returned in- or excluded?).

- The trajectory analyses needed long enough follow-up with no gaps in employment, i.e. with entitlement to use the occupational health care services. The descriptive characteristics of excluded subjects are presented in web-appendixes 1 and 2. Those who had 4 consecutive years of employment between 04/06/2004 and 04/19/2013 were included, and the follow-up started from their first, at least 4 years long employment period.

We have now added the following statement to the limitations: Moreover, excluding those with employment record for less than four years (n= 12,512) due to the need of long enough follow-up time reduced particularly the number of youngest employees (web-appendixes 1 and 2). (page 15)

It is unclear when age is measured; is that at the beginning of the follow-up period of an employee? I am not sure whether categorisation of age is appropriate as, for example, someone aged 24 will only be for the first year of the follow up period in the first category and for the other three years in the second age category.

- Age is measured at the beginning of the follow-up and the person remains in that group. We have clarified this in the text: Age was measured at the beginning of the follow-up and categorized into three groups: 20-24, 25-29 and 30-34-year-olds. (page 7)

Also, when following employees for 4 years or more one might expect promotions or switches between occupational class groups, e.g. semi-prof and managers? All these points should be discussed in the Method and/or Discussion section.

- We added this notification to the methodological considerations: A further limitation is that the initial occupational classes might have changed during the follow-up due to promotion or other changes in the employment. (page 15)

Within the Result section the mean days of follow-up is 1341 for men and 1328 for women. If employees are followed for 4 years or more one might expect a higher mean.

- Thank you for noticing this. This results from the inclusion criteria of the participants that slightly deviates from the strict 4 years criteria. This is due to fact that the young employees quite often face employment gaps between their temporary contracts. We have revised the inclusion criteria to be more precise. (page 6). As a note, the results with more strict time criteria (and less participants) remained practically identical while the selected criteria offered better power for the statistical estimates.

It is stated in the Result section that belonging to the high/recurrent trajectory was highest among manual workers, followed by routine and non-manual workers and semi-professionals. The text should include that this is compared to managers and no OHS visits. Furthermore, the presented 95%

confidence intervals don't suggest such a ranking. To present such as ranking, the authors should change the reference group to see, for example, whether semi-professionals (new reference group) are less likely in the high trajectory compared to routine workers. Besides, of more interest might be which occupational class is more likely to be in the high/decrease vs high/recurrent trajectory, and similarly no OHS visits vs low/increase trajectory.

- Thank you for these remarks. We have revised the text to state that in the case of occupational class, the reference category is managers. To further clarify, we have now mentioned in the Methods section that the trajectory-group is treated as the outcome in these analyses. (page 8).

- The results from multinomial logistic regression on occupational class with reversed order of classes are found in web-appendix 3. We have added a reference to these results in the sensitivity analyses (page 11).

- Other types of trajectory comparisons can be regarded as an important topic for future research.

Within the discussion it is expressed that this system is unique for Finland, Still, the authors should do their best to discuss and highlight what academics, doctors, and policy makers outside Finland could learn from the study.

- Thank you for this comment. We have now discussed this as follows: Our results show several important points for policy makers, as well as occupational and primary health care personnel, not only in Finland but also in countries with different primary care and occupational health care systems. According to our results preventive measures should be considered particularly among the trajectory groups of "Low/increasing" and "High/recurrent" health care utilization. In addition, special attention should be paid to the lowest occupational classes, and their OHS utilization should be closely monitored by the occupational health care/primary care in order to identify those in need for extra support. Case management protocols should be further developed and resources targeted in order to develop and maintain the health care system where early support is been given to those identified being in risk for subsequent work disability. As the preventive measures are done in practice, research should follow their success and produce evidence based development suggestions. In addition, OHS and primary care utilization requires more longitudinal research in order to target resources and preventive measures. (page 17)